# Effect of Ultraviolet Radiation and Benzo[a]pyrene Co-Exposure on Skin Biology: Autophagy as a Potential Target

**DOI:** 10.3390/ijms24065863

**Published:** 2023-03-20

**Authors:** Mohammad Fayyad-Kazan, Farah Kobaisi, Ali Nasrallah, Patrick Matarrese, Richard Fitoussi, Sandrine Bourgoin-Voillard, Michel Seve, Walid Rachidi

**Affiliations:** 1Department of Natural and Applied Sciences, College of Arts and Sciences, The American University of Iraq-Baghdad (AUIB), Baghdad 10001, Iraq; 2Univ. Grenoble Alpes, CEA, INSERM, IRIG-BGE UA13, 38000 Grenoble, France; 3Laboratoires Clarins, Centre de Recherche, 95000 Pontoise, France; 4Univ. Grenoble Alpes, CNRS, UMR 5525, VetAgro Sup, Grenoble INP, TIMC, 38000 Grenoble, France

**Keywords:** skin, ultraviolet radiation, benzo[a]pyrene, proteomics, autophagy

## Abstract

The skin is the outermost protective barrier of the human body. Its role is to protect against different physical, chemical, biological and environmental stressors. The vast majority of studies have focused on investigating the effects of single environmental stressors on skin homeostasis and the induction of several skin disorders, such as cancer or ageing. On the other hand, much fewer studies have explored the consequences of the co-exposure of skin cells to two or more stressors simultaneously, which is much more realistic. In the present study, we investigated, using mass-spectrometry-based proteomic analysis, the dysregulated biological functions in skin explants after their co-exposure to ultraviolet radiation (UV) and benzo[a]pyrene (BaP). We observed that several biological processes were dysregulated, among which autophagy appeared to be significantly downregulated. Furthermore, immunohistochemistry analysis was carried out to validate the downregulation of the autophagy process further. Altogether, the output of this study provides an insight into the biological responses of skin to combined exposure to UV + BaP and highlights autophagy as a potential target that might be considered in the future as a novel candidate for pharmacological intervention under such stress conditions.

## 1. Introduction

Skin, the body’s largest and outermost organ, comprises two primary layers: the epidermis (upper layer) and dermis (inner layer). The epidermis consists of several cell types, where keratinocytes are the predominant ones. On the other hand, the dermis includes mainly fibroblasts, nerves, blood vessels and sweat glands. Skin plays many vital roles, including protection against environmental insults (such as pathogenic microorganisms, exogenous chemicals and radiations), thermal regulation and synthesis of vitamin D, as well as sensory and immune functions [1]. Maintaining intact skin is, therefore, crucial for maintaining a healthy body [2,3]. Human skin, mainly the outer epidermis, is continuously exposed to environmental stressors, which can significantly impair, via oxidative stress, the main skin macromolecules. These modified macromolecules trigger different skin disorders, including skin ageing, psoriasis, inflammation or skin cancers [4,5]. Ultraviolet radiation (UVR) and polycyclic aromatic hydrocarbons (PAHs) such as Benzo[a]Pyrene (BaP) are widespread pollutants that could impact skin homeostasis [4,5,6]. The solar UVR spectrum consists of three sub-categories: UVA (320–400 nm), UVB (290–320 nm) and UVC (200–290 nm). UVR reaching the earth’s surface is composed mainly of UVA (90–95%) and UVB (1–5%), whereas most of UVC is absorbed by the ozone layer [7]. The biological effect of UVR on the skin varies according to the UV wavelength. Exposure to UVA has been associated with skin photo-aging [7,8,9], whereas UVB alone could account for sunburn [9,10]. UVA, along with UVB, could trigger photo-immunosuppression and the development of different cutaneous cancers (photo-carcinogenesis) [4,9]. Moreover, combining UVA with air pollutants, such as PAHs, has significantly worsened skin damage [11]. PAHs, a class of hydrocarbons composed of multiple benzene rings, are among the environment’s most abundant and toxic organic pollutants. Benzo[a]Pyrene (BaP), a highly mutagenic and carcinogenic PAH, is the most extensively studied member of this family of compounds. The abundance of this pollutant in the atmosphere is due to the fact that it is a key component of smoke released from wood burning, fuel consumption and cigarettes [5]. It may also be found in food due to the incomplete combustion of organic matter in grilled/smoked food [12,13]. BaP can enter the body through inhalation, ingestion or via oral and/or dermal routes. Following its metabolism, mutagenic reactive metabolites are produced, which might induce undesirable toxicity at the level of tissues and organs [14]. Exposure to BaP could trigger different pathologies, including hepatotoxicity, neurotoxicity, immunotoxicity and placental toxicity. Moreover, the literature has reported a link between mutations triggered by BaP and the development of cancer in various organs, including the lung, prostate, bladder and breast [15,16].

To date, the need for more information regarding the dysregulated biological processes of the human skin following BaP + UVR co-exposure urges detailed analysis at the proteomic level. In this study, we used mass-spectrometry-based proteomic analysis to identify differentially expressed proteins in human epidermal skin explants exposed to either no stress, UVR alone or UVR + BaP. With the help of pathway analysis, the dysregulated proteins were grouped into different pathways found to be modified post-stress exposure. Among the identified dysregulated pathways, autophagy appeared to be down-regulated. This was further validated using immunohistochemistry analysis. This observation highlights autophagy as being a potential target for the effects of UVR + BaP and a possible candidate for pharmacological treatments.

## 2. Results

### 2.1. Identification of Dysregulated Proteins following Skin Exposure to UV Alone or UV + BaP

In order to obtain an insight into the molecular alterations triggered in epidermal skin cells in response to UV irradiation coupled with BaP, comparative proteomic analysis was performed on skin explants following exposure to either nothing (control), UV radiation alone or UV + BaP. Proteins were extracted from different cellular fractions of the epidermis (cytosol, nucleus and cell membrane), followed by a FASP II sample preparation and LC-MS/MS analysis based on a label-free quantitative approach as described in the Materials and Methods. Bioinformatics analysis using Mascot and Paragon search engines enabled the identification of proteins dysregulated in the UV/control (1487 proteins in cytosolic fraction; 1387 proteins in the cell membrane fraction; 1516 proteins in the nuclear fraction, Appendix A) and UV + BaP/control (1521 proteins in cytosolic fraction; 1914 proteins in the cell membrane fraction; 1552 proteins in the nuclear fraction; Appendix A) conditions. Of the identified proteins, we only considered proteins with absolute ratios ≥1.5 (upregulated) and ≤0.67 (downregulated). Based on these criteria, we built a heatmap to visualize the protein expression patterns for the dysregulated proteins in the cytosolic, membrane and nuclear fractions (Figure 1A, 1B and 1C, respectively) upon double exposure vs. single exposure compared to the control (no exposure). Interestingly, depending on the sub-cellular fraction, the over-expression or under-expression of several proteins was exacerbated upon a double exposure (UV + BaP) compared to a single exposure to UV. In the case of the cytosolic fraction, 132 (24 upregulated and 108 downregulated) versus 319 (197 upregulated and 122 downregulated) proteins were dysregulated in the UV/control versus UV + BaP/control conditions (Figure 1A). For the cell membrane fraction, 86 (33 upregulated and 53 downregulated) versus 220 (130 upregulated and 90 downregulated) proteins were differentially expressed in the UV/control versus UV + BaP/control conditions (Figure 1B). In the case of nuclear fraction, 220 (85 upregulated and 135 downregulated) versus 216 (164 upregulated and 52 downregulated) proteins showed altered expression in the UV/control versus UV + BaP/control conditions (Figure 1C).

### 2.2. Identification of Biological Processes Dysregulated Following Skin Exposure to UV + BaP Compared to Exposition to UV Alone

From the complete lists of proteins identified and quantified in cytosol, cell membrane and nuclear extracts following UV alone or UV + BaP treatments, 147 proteins were selected for playing an important or specific role in skin biology. We analyzed the biological processes involving this subset of proteins and retained the top 10 biological processes that were dysregulated in UV + BaP-treated cells compared to UV alone (Table 1). The principal dysregulated processes are presented as a heatmap (Figure 2). Some processes are downregulated, including epidermis development, keratinocyte differentiation, keratinization, autophagy, negative regulation of endopeptidase activity and oxidation–reduction processes. Other processes, such as cell proliferation, collagen biosynthetic processes, viral processes and extracellular matrix organization, were upregulated. For some processes (such as the regulation of apoptosis, signal transduction, immune responses, cell–cell adhesion, the metabolic processes or the filament organization) the effects were less clear, whereby some proteins were upregulated while others were downregulated.

Intriguingly, among the dysregulated processes, autophagy was downregulated. This is of particular interest because the link between autophagy and skin biology is still elusive. Furthermore, attempts to link autophagy and UV irradiation were mainly based on identifying the common regulators induced by UV irradiation that might play a role in mediating autophagy with no thorough experimental studies, especially at the level of skin explants rather than individual skin cells [17]. Hence, in the upcoming experiments, we decided to focus on the involvement of autophagy in regulating skin biology in response to UV + BaP exposition.

### 2.3. Autophagy Is Downregulated after Co-Exposure of Skin Cells to UV + BaP

Autophagy is an intracellular digestion system during which cytosolic proteins or cellular organelles are degraded in response to certain environmental conditions, such as nutrient starvation. Autophagy can be examined via different approaches, and among which one is monitoring the localization of LC3 and its homologs. During autophagy, the cytoplasmic LC3-I protein becomes lipidated and conjugated to phosphatidylethanolamine (PE) to form LC3-II, which in turn is localized on the surface of nascent autophagosomes where the cargo is degraded. Since LC3-II is almost specifically associated with autophagosomes, monitoring LC3-II expression can supply information on the autophagic stage [18].

Quantifying LC3-II positive puncta via immunohistochemistry represents a gold-standard approach for assessing cellular autophagic activity [19]. Here, we observed that the number of LC3-II-positive puncta was higher in UV- or BaP-treated samples compared to control samples. Interestingly, the number of LC3-II-positive puncta was significantly reduced in UV + BaP-treated samples compared to UV- or Bap-treated samples (Figure 3A). However, the latter is not enough to conclude the effect of treatments on autophagy as LC3-II increase can be a marker of either increased autophagy or the blockade of the autophagosome proteosomal clearance due to the inhibition of autophagy. Due to this, and due to the discrepancy in the LC3-II expression profile, another molecular marker was also quantified: SQSTM1/P62. P62 is usually incorporated into the growing autophagosome as a cargo receptor where its degradation is a marker for active autophagy. The quantification of SQSTM1/P62 after the different forms of treatments was carried out using immunohistochemistry. All samples showed an increase in P62 staining compared to the untreated controls, signifying the failure of the elimination of aggregates and hence the autophagy blockade with UV irradiation mediating the highest inhibition (Figure 3B).

## 3. Discussion

In this study, we characterized, for the first time, the simultaneous effect of UV + BaP on skin explants. Intriguingly, we identified a functional dysregulation of different biological processes, particularly autophagy in the epidermis.

The skin represents the major defensive barrier of our body against environmental stressors such as ultraviolet radiation, air pollution, smoke and others. These environmental factors could lead to critical skin disorders, including aging and cancer. For instance, sunlight has already been demonstrated to trigger oxidative stress, photo-aging and photo-carcinogenesis [20]. Interestingly, air pollutants (such as particulate matter, nitrogen dioxide, sulfur dioxide, black carbon and carbon monoxide) are of significant interest for exerting a potential synergistic effect when simultaneously combined with UV radiation [11]. Due to the scarcity of studies assessing the combined effects of UV + BaP, we assessed, here, changes in the proteomic profile following the exposure of skin cells to UV + BaP. We identified hundreds of dysregulated proteins involved in several biological processes. The downregulated processes included epidermis development, keratinocyte differentiation, keratinization, autophagy, negative regulation of endopeptidase activity and oxidation–reduction processes, whereas the upregulated ones included proliferation, collagen biosynthetic processes, viral processes and extracellular matrix organization.

Exposure to UV radiation has been reported as the major risk factor triggering the so-called “extrinsic skin aging” [5]. Intriguingly, increasing evidence in the literature is now highlighting critical roles for air pollutants (such as ozone (O_3_) and particulate matter (PM)) in inducing skin damage and further exacerbating UV-induced skin disorders [10,21]. For instance, a recent study reported an additive effect of combining O_3_ and PM with UV leading to skin damage induction via a process called “oxInflammation”. Remarkably, it has also been reported that BaP, even at low concentrations, in combination with UVA, leads to enhanced oxidative damage, thus exacerbating extrinsic skin ageing and tumorigenicity [11]. In this context, several biological processes underlying skin normal functioning were impaired following UV + BaP exposure. For instance, epidermis development, keratinization and keratinocyte differentiation were downregulated. Several studies have previously examined the effect of UV on keratinocyte behavior and reported impaired proliferation and differentiation [22,23,24]. On the other hand, BaP has been reported to inhibit terminal differentiation and alter the growth of human epidermal keratinocytes [25,26]. In addition, we also observed a dysregulated redox balance, endopeptidase activity and extracellular matrix organization. These processes are well reported to be altered in several human skin disorders following exposure to UV radiation or air pollutants [27,28,29,30].

Remarkably, in this study, we observed that autophagy was downregulated following co-exposure to UV + BaP. Autophagy is a membrane-system-based cellular process that ensures either sustainable degradation and the clearing of damaged cellular components or the recycling of the cytoplasm’s content to maintain cell survival during starvation or other stress conditions such as UV radiation. Given its importance in maintaining homeostasis and a balanced cell life, autophagy is thus required to promote cell survival and longevity, where its activity drops with aging. It is well established in the literature that exposure to UV light or air pollutants such as BaP is a major factor driving skin aging via increasing the amounts of reactive oxygen species (ROS) levels which could damage skin macromolecules [17,31,32,33,34,35]. Impaired autophagy has been associated with several biological dysfunctions that can exacerbate the aging process, whereas induced autophagy generally triggers homeostasis and prolongs the cellular lifespan, thus accounting for healthy skin [36]. Observing that skin cells failed to induce autophagy upon co-exposure to UV + BaP could be of great interest, suggesting that such a combined treatment would accelerate skin aging in the absence of autophagy-mediated protection. Hence, pharmacological intervention to rescue autophagy under such conditions might be helpful to alleviate the resulting disorders such as aging. A detailed mechanistic understanding of the dysregulated autophagy function coupled with pharmacological screening to restore normal autophagy function would be indispensable in future studies.

## 4. Materials and Methods

### 4.1. Human Skin Explant Samples’ Exposition

Three abdominal skin explants (phototype I–II) were obtained from patients following the approved consent form, and cultured in media consisting of 2/3 Iscove-modified Dulbecco media (IMDM) complete media and 1/3 keratinocyte media with P/S and 5%FBS (Thermo Fisher Scientific, Waltham, MA, USA). The explants were either untreated, exposed to only UV irradiation (UVA 8 J/cm^2^ and UVB 10 mJ/cm^2^) [37] or exposed to benzo[a]pyrene pollutant (Sigma-Aldrich, Saint-Louis, Missouri, USA, B1760) at 100 nM in combination with UV irradiation. The doses of UV utilized were well established in our lab to initiate premature ageing. The irradiation was carried out using a BIO-SUN irradiation system (Vilber Lourmat, Eberhardzell, Germany). This was fitted out with a UVA irradiation source (365 nm) composed of T-20.L-365 tubes (no UVB, no UVC emissions), mercury vapor tubes, low pressure and hot cathodes. Then, the stimulator was fitted out with a UVB irradiation source (312 nm) composed of T-15.M-312 tubes (no UVA, no UVC emission). The radiometer was linked to an energy-programmable microprocessor allowing for the adjustment of the duration and the energy received by the skin explants. These UV irradiations were sufficient to induce reproducible alterations in the epidermis and dermis. The treatment schedule was on Days 1, 2, 5 and 6. On Day 7, skin explants were collected, and their epidermis was isolated for further analysis.

### 4.2. Proteomic Samples’ Processing

All reagents were purchased from Sigma-Aldrich unless otherwise specified. On Day 7, after the different treatment schedules, the samples were collected with the separation of the dermis and epidermis, where the latter was stored at -80°C. Samples were dilacerated and homogenized in liquid nitrogen. Homogenates were incubated in 0.1 M Tris-HCL pH 8.5 buffer containing 2% SDS and 0.1 M DTT during two periods of 5 min at 95 °C each, and in between which, freezing was applied for 15 min at −80 °C. Samples were sonicated for 1 min per 3 s pulses, with each pulse followed by ice cooling. The resulting lysates were centrifuged (15,000× *g*, 10 min), and supernatants were collected. Proteins were extracted sequentially to preferentially isolate cytosolic proteins, nuclear proteins and membrane proteins from isolated epidermis (from skin explant) following a commercially available protein extraction kit (ProteoExtract^®^ Subcellular Proteome Extraction Kit, Millipore, Fontenay sous Bois, France).

Proteins were precipitated in acetone/TCA (ratio 9:1) at −80 °C for 2 h. Precipitated proteins were collected via centrifugation (20,000× *g*, 20 min, 4 °C), and pellets were washed twice in an acetone/methanol/acetic acid ratio (8:1:0.02).

Precipitated proteins were dissolved in Tris-HCl 0.1M buffer containing 8M urea and 2% DMSO. The protein content was estimated via BCA assay (Pierce; Thermo Scientific, Rockford, IL, USA).

The protein extract was digested according to the FASP II procedure as described by Wiśniewski et al. [1], with slight modifications. The protein extracts (500 µg) were loaded onto Amicon Ultra 0.5 10 kDa filters (Millipore, Billerica, MA, USA), and filters were washed with 8 M urea in 0.1 M Tris (pH 8.5) to facilitate the removal of protein-bound small molecules. Proteins were reduced with 10 mM DTT (dithiothreitol) (Euromedex, Souffelweyersheim, France) for 60 min and alkylated with 50 mM iodoacetamide for 30 min. Urea buffer was exchanged via ultrafiltration with ammonium bicarbonate 50 mM. Cleaned proteins were digested overnight on the filter with trypsin at a 1:50 ratio (Promega, Madison, WI, USA), and the resulting peptides were released from the filter via centrifugation (14,000× *g*, 20 min) followed by a first wash of the filter in 0.5 M NaCl and second wash of the filter with 50% acetonitrile. All three fractions were pooled, and peptides were diluted at 5% acetonitrile and acidified with 0.1% trifluoroacetic acid at a 0.1% final concentration. Peptides were desalted using 500 mg SepPak tC18 sample extraction columns (Waters, Milford, DE, USA), eluted with 1.2 mL of 70% acetonitrile, dried and dissolved in 200 µL of 5% acetonitrile acidified with 0.1% formic acid.

### 4.3. LC-MS/MS Data Acquisition and Analysis

Peptide digests (126 ng per run) were loaded onto a nanoACQUITY UPLC Symmetry C18 Trap Column, 180 µm × 20 mm (particle diameter 5 μm, pore size 100 Å) in trap and elute mode with ACQUITY UPLC Peptide BEH C18 nanoACQUITY Column 75 µm × 250 mm (particle diameter 1.7 μm, pore size 130 Å) (Both Waters, Milford). Run gradient was performed using the Eksigent Ultra Plus nano-LC 2D HPLC (Sciex, Framingham, MA, USA) system over 90 min with a gradient from 3% to 40% buffer B (buffer A: 0.1% formic acid; buffer B: 95% acetonitrile, 0.1% formic acid) at a flow rate of 300 nl/min. The Eksigent system was coupled to a TripleTOF^®^ 5600 (Sciex, Framingham) mass spectrometer interfaced to a nanospray III source, whereby parameters were set as follows: IS at 2500 V, curtain gas at 30 psi, Gas Sprayer 1 at 1 psi and interface heater temperature at 150 °C. Acquisition parameters were as follows: for DDA (data-dependent acquisition) mode, one 250 ms MS scan (>30 K resolution). Following each survey MS1 scan, MS/MS spectra for the 30 most abundant parent ions (*m*/*z* range 350–1250) were acquired (high sensitivity mode, >15 K resolution). For DIA (data-independent acquisition) mode, it was one 150 ms MS scan (>30 K resolution), followed by 35 fixed SWATH windows each with a 75 ms accumulation time and a 350–1250 *m*/*z* range. MS/MS SWATH scans (high sensitivity mode, >15 K resolution) were set at a 26 amu window, with Q1 isolation windows covering the entire mass range.

DDA spectra processing and database searching were performed with ProteinPilot (v4.5 beta, Sciex, Framingham) using the Paragon algorithm. The search parameters were as follows: sample type: identification; cys-alkylation: iodoacetamide; digestion: trypsin; instrument: TripleTOF 5600; special factors: urea denaturation; ID focus: biological modifications. The database was downloaded from Uniprot (June 2014), filtering for reviewed human proteins only (20,194 entries). The resulting group file was loaded into Peakview^®^ (v2.0, Sciex, Framingham), and peaks from the SWATH runs were extracted with a peptide confidence threshold of 99% and a false discovery rate <1%. Label-free quantification was performed using Marker View (v1.2.1, Sciex, Framingham). The selection of the proper peak was performed using the automated assistance of PeakView. Absolute signals of peptides and proteins were calculated by summing the extracted area of all unique fragment ions.

### 4.4. Bioinformatic Analysis

Among the identified proteins, we selected for further analysis those that both passed the Student test (*p* < 0.05) and whose fold changes were ≥2 (up-regulated proteins) or ≤0.5 (down-regulated proteins). The enrichment analysis for gene ontology terms (biological processes, molecular functions or cell components) was conducted using DAVID functional annotation analysis (Bioinformatics Resource v6.8, https://david.ncifcrf.gov/tools.jsp, accessed on 20 November 2018) by setting the human genome as the background [38,39].

Enrichment analysis works by (1) building associations between identified proteins from the MS assay whose expressions are considered to be modulated because of the differential conditions, and the terms describing their biological, cellular or molecular functions (GO annotations, gene ontology), and the metabolic pathways to which they belong, (2) grouping proteins associated with the same descriptive terms and (3) looking for the significance of these protein groups/descriptive terms and associations through statistical tests. The enrichment analysis used was Fisher’s exact test, and the multiple testing procedure was Bonferroni’s correction, as it is the most conservative one. Results were considered to be significant for a *p*-value inferior or equal to 0.05.

### 4.5. Immunohistochemistry Analysis

On Day 7, following the different treatments stated above, samples were fixed in formalin solution (Sigma, HT50-1-1) for 48 to 72 h. After fixation, the explants were embedded in paraffin, and slices of 5 µm of thickness were studied. The slices were stained with primary antibodies against LC3B (ab192890, ABCAM, Cambridge, UK) or SQSTM1 (Abcam, ab109012), followed by a secondary antibody conjugated with Alexa Fluor 488 dye (A11008, Life technologies, Carlsbad, CA, USA). Seventeen to twenty pictures per condition were acquired via fluorescence microscopy, and the intensity of staining in the epidermis was analyzed using proprietary methods in Visilog software. Wilcoxon’s test was used to analyze the results, and a *p*-value lower than 0.05 indicated a statistically significant difference.

### 4.6. Statistical Analysis

Statistical analysis was carried out using R Studio (Wilcoxon test).

## 5. Conclusions

In conclusion, this study brings about new insights regarding the molecular consequences of skin co-exposure to UV + BaP. It also highlights the important role of autophagy that could be regarded as a potential target for pharmacological treatment.

## Figures and Tables

**Figure 1 ijms-24-05863-f001:**
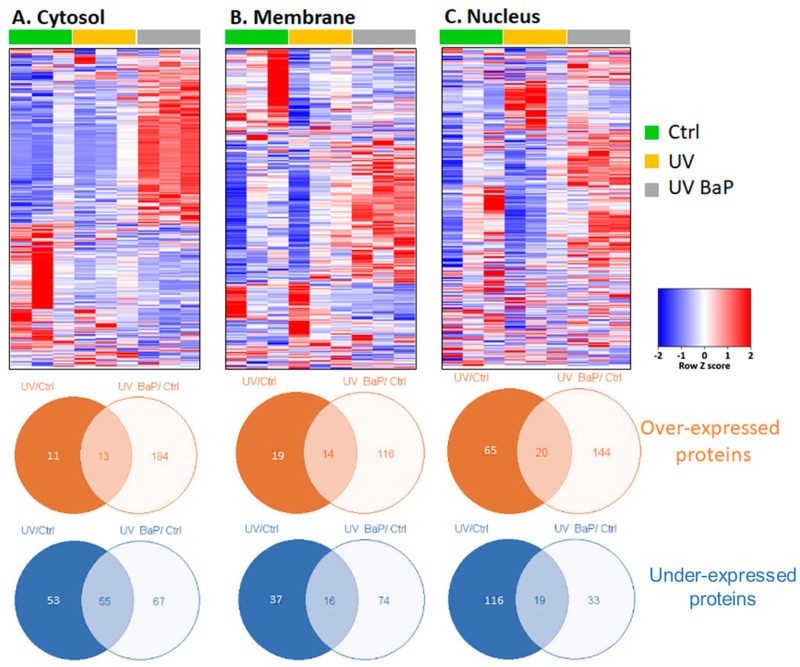
Dysregulation of proteins from the cytoplasmic (**A**), cell membrane (**B**) and nuclear (**C**) extracts of human skin explants exposed to UV light or UV light and BaP, compared to untreated cells (Crtl). Each heatmap represents the expression profile of dysregulated proteins upon UV exposure with or without a co-exposure to BaP, based on the z-score of the protein’s normalized peak area. Rows represent dysregulated proteins and columns represent the samples at different exposure conditions. The red color represents an overexpression while the blue color represents a down-regulation of the protein compared to the mean value of a protein from all samples. The z-score is a normalization score of each row and has the same range of color values. Heatmaps were built using the Funrich tool, version 3.1.3. For each Venn diagram, the number of proteins over-expressed (*p* < 0.05; ≥1.5 fold) or under-expressed (*p* < 0.05; ≤0.67 fold) is indicated for each sub-localization of cells.

**Figure 2 ijms-24-05863-f002:**
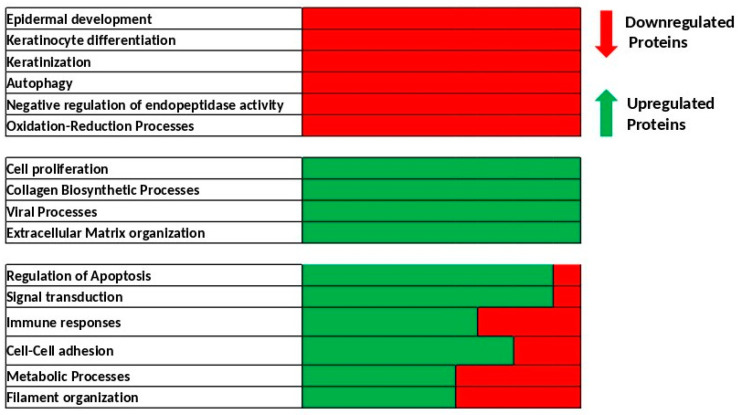
The major upregulated and downregulated biological processes following UV + BaP co-exposure. The heat map displays the major dysregulated biological process and the distribution of the identified proteins either upregulated (green color) or downregulated (red color), among each particular process. Some deregulated pathways solely harbor proteins that were all either downregulated (such as autophagy) or upregulated (such as cell proliferation) in the proteomics data, while other pathways have some of the proteins involved in these processes upregulated and others are downregulated (such as cell–cell adhesion).

**Figure 3 ijms-24-05863-f003:**
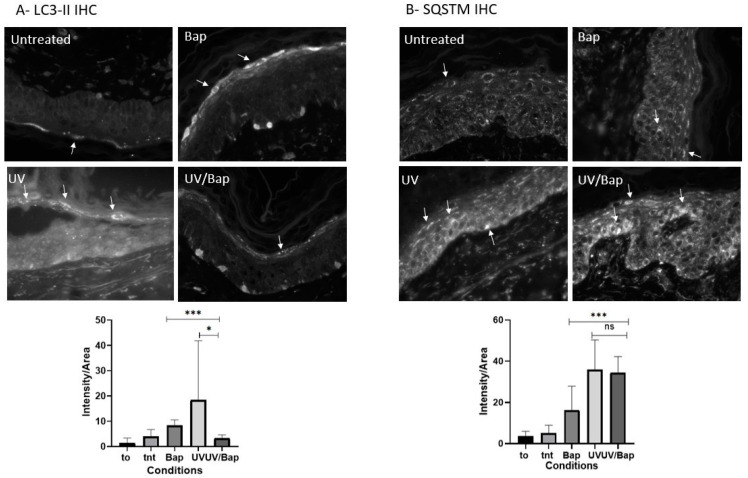
Immunohistochemistry of skin explants for autophagy markers LC3-II and SQSTM. Seven days post the different treatments, either sham treated or exposed to UV and/or Bap, the staining of autophagy markers was conducted on the explants. (**A**) LC3-II staining; an increase in LC3-II puncta was detected upon the exposure of the samples to either UV or Bap compared to the untreated samples. Moreover, an increase was also detected for samples treated with both UV and Bap, but this was less prominent when compared to each treatment alone. (**B**) SQSTM staining; all of the different treatments displayed an increase in SQSTM staining, signifying a failure to eliminate autophagosomal aggregates. Examples of puncta are pointed out by arrows. To: explant on Day 0, tnt: untreated, Bap: Benzo[a]pyrene exposed, UV: UV irradiated, UV/Bap: UV irradiated and Benzo[a]pyrene exposed explant, *: *p* < 0.05, ***: *p* < 0.001, ns: non significant, 400× magnification (Obj. 40×).

**Table 1 ijms-24-05863-t001:** List of the dysregulated biological processes following UV + BaP co-exposure compared to UV exposure alone. The top 10 dysregulated biological processes are indicated for upregulated or downregulated proteins from cytosolic, cell membrane and nuclear extracts. Count indicates the number of proteins involved in each biological process.

Terms	Protein Count	*p*-Value
Biological processes of cytosolic upregulated proteins specific to UV + BaP condition		
SRP-dependent cotranslational protein targeting to membrane	66	5.80 × 10^−111^
viral transcription	66	3.20 × 10^−103^
nuclear-transcribed mRNA catabolic process, nonsense-mediated decay	66	9.50 × 10^−101^
translational initiation	67	2.10 × 10^−97^
rRNA processing	68	2.70 × 10^−83^
translation	69	2.10 × 10^−79^
cytoplasmic translation	14	5.90 × 10^−20^
ribosomal large subunit assembly	8	1.30 × 10^−09^
ribosomal small subunit biogenesis	7	9.20 × 10^−09^
ribosomal small subunit assembly	7	3.10 × 10^−08^
Biological processes of cytosolic downregulated proteins specific to UV + BaP condition		
negative regulation of endopeptidase activity	7	7.20 × 10^−06^
epidermis development	6	1.90 × 10^−05^
negative regulation of peptidase activity	4	3.60 × 10^−05^
keratinocyte differentiation	5	2.10 × 10^−04^
negative regulation of cytokine-mediated signaling pathway	3	5.20 × 10^−04^
keratinization	4	8.50 × 10^−04^
peptide cross-linking	4	9.50 × 10^−04^
innate immune response	8	1.30 × 10^−03^
negative regulation of catalytic activity	4	3.10 × 10^−03^
cell–cell adhesion	6	4.00 × 10^−03^
Biological processes of cell membrane upregulated proteins specific to UV + BaP condition		
viral entry into host cell	6	1.30 × 10^−04^
protein transport	9	3.00 × 10^−03^
substantia nigra development	4	3.40 × 10^−03^
small GTPase-mediated signal transduction	7	4.10 × 10^−03^
leukocyte migration	5	6.70 × 10^−03^
extracellular matrix organization	6	7.20 × 10^−03^
vesicle-mediated transport	5	1.40 × 10^−02^
antigen processing and presentation of peptide antigen via MHC class I	3	1.40 × 10^−02^
membrane organization	3	1.60 × 10^−02^
regulation of oxidative-stress-induced intrinsic apoptotic signaling pathway	2	1.80 × 10^−02^
Biological processes of cell membrane downregulated proteins specific to UV + BaP condition		
tricarboxylic acid cycle	5	6.80 × 10^−06^
keratinocyte differentiation	5	3.20 × 10^−04^
keratinization	4	1.10 × 10^−03^
peptide cross-linking	4	1.30 × 10^−03^
2-oxoglutarate metabolic process	3	2.70 × 10^−03^
epidermis development	4	5.80 × 10^−03^
neutrophil aggregation	2	8.60 × 10^−03^
chemokine production	2	1.30 × 10^−02^
sequestering of zinc ion	2	1.70 × 10^−02^
ketone body catabolic process	2	1.70 × 10^−02^
Biological processes of nucleus upregulated proteins specific to UV + BaP condition		
positive regulation of telomerase RNA localization to Cajal body	8	1.60 × 10^−11^
positive regulation of establishment of protein localization to telomere	7	2.70 × 10^−11^
positive regulation of protein localization to Cajal body	6	2.20 × 10^−09^
positive regulation of telomere maintenance via telomerase	7	2.50 × 10^−07^
response to drug	12	5.60 × 10^−05^
cell–cell adhesion	11	1.00 × 10^−04^
protein folding	9	1.50 × 10^−04^
protein stabilization	8	1.60 × 10^−04^
binding of sperm to zona pellucida	5	2.10 × 10^−04^
toxin transport	5	2.30 × 10^−04^
Biological processes of nucleus downregulated proteins specific to UV + BaP condition		
autophagy	5	6.60 × 10^−05^
activation of cysteine-type endopeptidase activity involved in apoptotic process	4	3.50 × 10^−04^
neutrophil aggregation	2	3.30 × 10^−03^
chemokine production	2	5.00 × 10^−03^
sequestering of zinc ion	2	6.70 × 10^−03^
positive regulation of peptide secretion	2	6.70 × 10^−03^
leukocyte migration involved in inflammatory response	2	1.80 × 10^−02^
positive regulation of NF-kappaB transcription factor activity	3	2.10 × 10^−02^
astrocyte development	2	2.60 × 10^−02^
regulation of cytoskeleton organization	2	3.10 × 10^−02^

## Data Availability

Data available on request due to privacy restrictions.

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
