# Peer review of "Effect of Ultraviolet Radiation and Benzo[a]pyrene Co-Exposure on Skin Biology: Autophagy as a Potential Target"

_ijms, 2023, doi:10.3390/ijms24065863_

Round 1

Reviewer 1 Report

In the manuscript titled "Effect of Ultraviolet radiation and Benzo[a]pyrene co-exposure on skin biology: Autophagy as a potential target," the authors Fayyad-Kazan M. et al., have studied the effect of UV radiation and polyaromatic hydrocarbons on skin explants. Their results demonstrate extensive proteomic changes in the skin explants exposed to different environmental stressors. They have further characterized the status of one of the dysregulated processes - autophagy. The significant comments on the manuscript are:

The authors have mentioned in the introduction that BaP is individually capable of causing various skin pathologies. However, the authors did not include this as one of the treatment groups for their proteomic analysis, nor were the data from a previous study that analyzed similar parameters used to compare the effect of co-exposure vs. single exposure. The authors should justify this. 

The authors have not provided information on the procedure used to segregate the cellular compartments mentioned. Also, the authors have said that they isolated cell membrane-associated proteins. However, they note the term membrane fraction in the rest of the manuscript. Since various cellular organelles have membranes, It would be helpful if the authors continue accurately describing the fraction of what it contains.

The results of section 2.1 need to be summarized to explain the global effect on protein dysregulation due to double exposure vs. single exposure and what type of dysregulation was predominant in every kind of fraction upon double exposure vs. single exposure. 

In lines 128-129, the authors mentioned that the link between autophagy and skin biology is not well explored. However, more compelling evidence would be literature that indicates autophagy is or could be involved in response to UVR or PAH exposure in other tissues, warranting a need to study if this is conserved in the skin.

According to Table 1, only five autophagy-related genes in the nuclear fraction are downregulated. As a result, the depiction in Figure 2 that autophagy is the 4th highest process which is overall downregulated, needs to be justified with the number of down-regulated genes of each function. 

In lines 152 - 153, the authors need to cite studies that make that statement or conclusion.

In section 2.3, the authors mention that they carried out SQSTM1 puncta quantification to overcome the limitation of LC3 puncta to reflect autophagic flux accurately. However, such a conclusion using IF can only be made using co-IF of both LC3 and SQSTM1, where there is an increase in the co-localized puncta. In the absence of such an assay, autophagosome - lysosome fusion inhibitors could also be used to demonstrate the effect of autophagic flux. Further, the authors need to justify why they see a decrease in levels of LC3 puncta in UV+BaP treated cells while SQSTM1 puncta are high.

In figures 3A and 3B, the authors need to highlight the areas they are referencing as LC3/SQSTM1 puncta. The figure and the text in section 2.3 mention the assay as IHC, but the figure legend says it as IF. The authors need to clarify this.

Reviewer 2 Report

Summary: This work uses abdominal skin explants treated either in control conditions, with UV, with BaP or with UV+BaP and uses mass-spectometry-based methods to quantify changes in protein expression 24hr after treatment. Using this proteomic and bioinformatics analysis the team compiled lists of proteins from cytosolic, membrane and nuclear fractions which were substantially upregulated or downregulated in UV+BaP/Ctl samples compared to UV/Ctl samples. Of the many cellular processes that were dysregulated in UV+BaP compared to UV alone, the authors focused on autophagy (downregulated in the nuclear fraction) to perform follow-up studies. Using IHC the authors examined two markers of autophagy in the skin: LC3-II and SQSTM1/p62. The authors conclude that the IHC confirms their proteomic findings and that autophagy blockade is therefore a potential target for future studies of combined environmental toxicants in the skin. The authors are commended for their use of ex-vivo skin and proteomic analysis of combinatorial effects, which definitely requires further study. The overall premise is sound but the manuscript would require revisions for publication.

1) English needs a little further polishing.

2) The methods do not specify how cytosolic, membrane or nuclear extracts were specifically made. Also, how many explants were used?

3) It is still unclear to this reviewer why there is such a large discrepancy between Fig 3A and Fig 3B. Why would UV+BaP block LC3-II so dramatically compared to UV alone? Maybe because the cells are now pushed to apoptosis or necrosis?

4) There are many interesting pathways that emerged from the proteomic analysis. It’s unfortunate that only one was analyzed by IHC. It would greatly strengthen the paper to include additional analyses. Markers of pathways such as the innate immune response, keratinization, or keratinocyte differentiation should be relatively straightforward to do.

5) The ratio of UVA+UVB described is not close to what skin would naturally see (which is usually about 90-95% UVA and 5-10% UVB). Authors should clarify the bulb system they used. How did they decide on the dose?

6) It’s unfortunate that the proteomic analysis did not include samples exposed to BaP alone, but maybe that was a budgetary constraint. Can the information in Figure 1 be expanded upon to reveal more information? For example, there were 13 proteins upregulated in common in the cytosolic fraction between the UV/Ctl and UV+BaP/Ctl groups. Are these proteins important to skin biology, stress response, apoptosis, proliferation, etc? Did treatment with BaP enhance or reduce their dysregulation compared to UV alone?

7) Figure 2 needs a little polishing. The arrows at the top are unclear and there is extra coloration between “extracellular matrix organization” and “regulation of apoptosis”.

Round 2

Reviewer 2 Report

English is much improved, as is one of the figures. Methods section is improved as well.

Maybe I missed it, but I would suggest citing the previous work that used the current dose of UV for skin ageing studies if that really isn't included here.

It is unfortunate that you are forgoing the opportunity to show additional data using the samples you currently have to strengthen your data confirming some of the other pathways that you found to be altered by treatment using proteomics. 

Author Response

-We thank the reviewer for his comments. The reference citing the used dose of UV for skin ageing trigger has been added following this comment. Please see revised version.

-Regarding the other identified pathways, hopefully more validation and understanding of their involvement in the skin ageing process will be carried out in our future research work and publications.